SciPost Physics

Submission

# Adiabatic gauge potential and integrability breaking with free fermions

Balázs Pozsgay[1*], Rustem Sharipov[2], Anastasiia Tiutiakina[3] and István Vona[1,4]

**1** MTA-ELTE "Momentum" Integrable Quantum Dynamics Research Group,
Department of Theoretical Physics, ELTE Eötvös Loránd University, Budapest, Hungary
**2** Physics Department, Faculty of Mathematics and Physics, University of Ljubljana,
Ljubljana, Slovenia
**3** Laboratoire de Physique Théorique et Modélisation, CNRS UMR 8089, CY Cergy
Paris Université, 95302 Cergy-Pontoise Cedex, France
**4** Holographic Quantum Field Theory Research Group, Institute for Particle and
Nuclear Physics, HUN-REN Wigner RCP, Budapest, Hungary
\* pozsgay.balazs@ttk.elte.hu

March 6, 2024

## Abstract

We revisit the problem of integrability breaking in free fermionic quantum spin chains. We investigate the so-called adiabatic gauge potential (AGP), which was recently proposed as an accurate probe of quantum chaos. We also study the so-called weak integrability breaking, which occurs if the dynamical effects of the perturbation do not appear at leading order in the perturbing parameter. A recent statement in the literature claimed that integrability breaking should generally lead to an exponential growth of the AGP norm with respect to the volume. However, afterwards it was found that weak integrability breaking is a counter-example, leading to a cross-over between polynomial and exponential growth. Here we show that in free fermionic systems the AGP norm always grows polynomially, if the perturbation is local with respect to the fermions, even if the perturbation strongly breaks integrability. As a by-product of our computations we also find, that in free fermionic spin chains there are operators which weakly break integrability, but which are not associated with known long range deformations.

# 1   Introduction

In quantum many body physics there is the dichotomy of chaos versus integrability. The two families of systems display markedly different dynamical behaviour. For example, isolated chaotic models equilibrate to the Gibbs ensemble [1], and they typically feature diffusive transport of their conserved quantities (such as energy or particle number). In contrast, integrable models equilibrate to the Generalized Gibbs Ensemble [2], and typically they support ballistic transport. Integrable models include free systems (in arbitrary space dimension), and also a plethora of interacting models in one space dimension.

Practically every quantum many body system can be uniquely characterized as either chaotic or integrable, at least in the thermodynamic limit, where both the system size and the number of particles is approaching infinity. However, for a finite size system it can be more difficult to decide between quantum chaos and integrability. This motivated the search for numerical probes of quantum chaos.

Traditionally there have been two main methods to treat this problem. First, if a given model is suspected to be integrable, one can attempt to obtain exact many body eigenstates using the method of the Bethe Ansatz [3]. Alternatively, one could try to find the higher conserved charges or the standard algebraic structures of integrability such as Lax operators and $R$-matrices [4]. If either one of these methods works, then the model is found to be integrable. However, these special analytic computations can be difficult to carry out, and often they require an expert working with one dimensional integrable models. The alternative and by now quite standard method is to investigate the level spacing statistics of the finite volume systems. Chaotic models display Wigner-Dyson statistics, while integrability leads to a Poisson distribution [5, 6]. In the case of spin chains exact diagonalization can be performed in a straightforward way, yielding direct information about the question at hand.

However, level spacing statistics might not be the optimal tool in all cases. In certain situations it is not possible or very demanding to reach large enough volumes for the method to work. For example, there are models which might be "close" to an integrable

model, either because they arise from small perturbation of a known model [7, 8], or perhaps simply by chance [9, 10]. In such cases the signs of quantum chaos become visible only in larger volumes, making then numerical work more challenging.

In the recent work [11] the so-called Adiabatic Gauge Potential (AGP) was proposed as an alternative probe of quantum chaos. The AGP carries information not only about the level spacings, but also about the matrix elements of local operators. The key statement of [11] was, that if a chaotic or integrable model is perturbed by a local operator, then the norm of the AGP behaves differently, depending on whether the original model was chaotic or integrable, and on whether the perturbation itself is along an integrable direction. It was argued that if the model is chaotic, or if the model is integrable but the perturbed model becomes chaotic, then the AGP scales exponentially with the volume. In contrast, if the model is integrable and the perturbation is also along an integrable direction, then the AGP scales polynomially.

Following [11] the method of AGP was also considered in the work [12], which treated the so-called "weak breaking of integrability". Here the description "weak" refers to a special selection of the perturbing operators, such that the integrability breaking effects do not arise to the leading orders in the perturbing parameters [13–15]. Most importantly, for these types of systems the thermalisation time is larger than usually, and it scales as $t_{th} \sim \lambda^{-\kappa}$ with $\kappa > 2$, as opposed to the standard case of $\kappa = 2$. Such a weak breaking was considered recently in multiple works [16–18]; it was explained in [17, 19] that weak integrability breaking is closely related to long range deformations known from the literature dealing with the AdS/CFT correspondence in the planar limit [20, 21]. It was a key statement of [12] that weak integrability breaking leads to a crossover between polynomial and exponential growth of the AGP, as the perturbation strength is increased.

In this paper we revisit the behaviour of the AGP in spin chains which are equivalent to free fermions. We show that the picture painted in the previous works is not complete: We argue that in free fermionic models the AGP scales polynomially even if the perturbation breaks integrability, given that the perturbation is local with respect to the fermions. The distinction between weak and strong integrability breaking is irrelevant in such situations. This observation about free fermionic models has escaped previous works dealing with this problem.

Regarding weak integrability breaking there is an open question: Does the framework of the long range deformations [20, 21] cover all cases of weak breaking? We also contribute to this question. In the case of free fermions we show that there are perturbations, which are local in space and also local with respect to the fermions, that are weakly breaking integrability, and that do not follow from the boost or bi-local long range deformations discussed in [19].

We should mention that there is some similarity of our computations and those in the recent work [22], which considered the perturbation of free fermions in the continuum. However, the primary goals and thus the results obtained are different.

## 2 Adiabatic gauge potential and integrability breaking

In this Section we set the stage for the computations to be presented in later Sections. First we introduce the key definitions and a few basic statements in the literature.

## 2.1 Adiabatic gauge potential

Consider a Hamiltonian $H(\lambda)$, which depends on a parameter $\lambda$. This Hamiltonian has a set of orthonormal eigenstates $|n(\lambda)\rangle$, satisfying the Schrödinger equation:

$$H(\lambda)|n(\lambda)\rangle = E_n(\lambda)|n(\lambda)\rangle. \tag{1}$$

For simplicity we assume that the spectrum is non-degenerate, so that every eigenvector is well defined up to a phase. Cases with degeneracies will be treated afterwads. Then it is always possible to define a unitary transformation that gradually rotates these eigenstates:

$$|n(\lambda)\rangle = U(\lambda)|n(0)\rangle. \tag{2}$$

The unitary transformation is well defined, up to the choice of the phases of the vectors.

The generator of this transformation is called the *adiabatic gauge potential*, denoted as $\mathcal{A}_\lambda$. The formal definition is:

$$\mathcal{A}_\lambda = i\left[\partial_\lambda U(\lambda)\right]U^\dagger(\lambda), \quad \mathcal{A}_\lambda|n(\lambda)\rangle = i\partial_\lambda|n(\lambda)\rangle. \tag{3}$$

There is some ambiguity in choosing the diagonal components of $\mathcal{A}_\lambda$ in the eigenbasis of $H(\lambda)$, because the relative phases of the eigenvectors could be chosen as an arbitrary function of $\lambda$. Here we adopt the standard definition that

$$\langle n(\lambda)|\mathcal{A}_\lambda|n(\lambda)\rangle \equiv 0. \tag{4}$$

It can be easily shown that the AGP satisfies the following operator equation [23, 24]

$$i\partial_\lambda H(\lambda) = \left[\mathcal{A}_\lambda, H(\lambda)\right] - i\mathcal{F}(\lambda), \tag{5}$$

where the operator $\mathcal{F}(\lambda)$ is diagonal in the eigenbasis of $H(\lambda)$. Explicitly, it is given by

$$\mathcal{F}(\lambda) = -\sum_n \frac{\partial E_n(\lambda)}{\partial\lambda}|n(\lambda)\rangle\langle n(\lambda)|. \tag{6}$$

The matrix elements of the AGP between the eigenstates of $H(\lambda)$ can be expressed as:

$$\langle m(\lambda)|\mathcal{A}_\lambda|n(\lambda)\rangle = -\frac{i}{\omega_{mn}}\langle m(\lambda)|\partial_\lambda H(\lambda)|n(\lambda)\rangle, \qquad \omega_{mn} = E_m(\lambda) - E_n(\lambda). \tag{7}$$

These definitions of the AGP are ill-defined if there are degeneracies in the spectrum. In the degenerate cases the unitary rotation $U(\lambda)$ is ill-defined, and it might happen that the derivatives in (3) become singular. This can be avoided by degenerate state perturbation theory, thus making sure that we always choose a basis in which the perturbation is diagonal within the degenerate subspaces. However, this would complicate both the formal and the numerical treatment.

Alternatively, it is possible to define a regularized form of the AGP, which is convenient for both purposes. Following [11] we define the regularized AGP through the matrix elements

$$\langle m(\lambda)|\mathcal{A}_\lambda(\mu)|n(\lambda)\rangle = -\frac{i\,\omega_{mn}}{\omega_{mn}^2 + \mu^2}\langle m(\lambda)|\partial_\lambda H(\lambda)|n(\lambda)\rangle, \tag{8}$$

where $\mu$ is a small energy cutoff. This regularized form can be expressed alternatively as

$$\mathcal{A}_\lambda(\mu) = -\frac{1}{2}\int\limits_{-\infty}^{\infty} \text{sign}(t)e^{-\mu|t|}(\partial_\lambda H)(t), \qquad (\partial_\lambda H)(t) \equiv e^{iHt}\partial_\lambda H e^{-iHt}. \tag{9}$$

In the next section we discuss the AGP as a probe of quantum chaos and explain the proper choice of the energy cutoff $\mu$.

## 2.2 AGP as a probe of quantum chaos

In the recent work [11] it was proposed that the AGP can serve as an accurate probe of quantum chaos. The key idea was to study the volume dependence of the norm of the AGP, which would tell the difference between chaos and integrability. Now we briefly summarize the main statements of [11].

We start with a perturbation

$$H(\lambda) = H_0 + \lambda V, \tag{10}$$

where $H_0$ can be chaotic or integrable, and if $H_0$ is integrable in itself, the perturbed model $H(\lambda)$ might or might not be integrable with a finite $\lambda$. For such a perturbation we can compute the norm of the AGP as

$$\|\mathcal{A}_\lambda\|^2 = \frac{1}{\mathcal{D}} \sum_n \sum_{m \neq n} |\langle m| \mathcal{A}_\lambda |n\rangle|^2, \tag{11}$$

with $\mathcal{D}$ being the dimension of the Hilbert space. The matrix elements of $\mathcal{A}_\lambda$ are given by (7) or by the regularized formula (8). For the regulator the proper choice was found to be $\mu \sim L\mathcal{D}^{-1}$ [11]. We use this regulator in all our numerical examples below.

The investigation of [11] revealed intriguing scaling behavior with respect to the system size $L$. If $H_0$ is chaotic, or if it is integrable but the perturbation breaks integrability, then the norm of the AGP exhibits exponential scaling:

$$\frac{\|\mathcal{A}_\lambda\|^2}{L} \sim e^{\kappa L}, \tag{12}$$

for any finite parameter $\lambda$. This scaling behaviour should hold in the $L \to \infty$ limit.

On the other hand, if $H_0$ and also $H(\lambda)$ are integrable, then a polynomial bound on the AGP norm was found:

$$\frac{\|\mathcal{A}_\lambda\|^2}{L} \sim L^p, \qquad p \in \mathbb{R}. \tag{13}$$

An intriguing aspect of the adiabatic gauge potential (AGP) is its behavior at $\lambda = 0$. It was claimed in [11] that in most cases the scaling should be the same for $\lambda = 0$ and $\lambda \neq 0$. This would make it possible to distinguish integrable models and integrable perturbations from all other cases, already at $\lambda = 0$. However, the situation is more complicated: there are exceptional cases when the scaling at $\lambda = 0$ differs from that at $\lambda \neq 0$. These are the cases that we treat in this paper.

First we focus on the cases with weak integrability breaking, and we review the known results regarding this class of perturbations. Afterwards in Section 3 we discuss that special cases that appear in free fermionic models; these are new results of the present work.

## 2.3 Weak integrability breaking

Let us consider the case when $H_0$ is integrable, but the perturbation in (10) is breaking integrability. We say that we are dealing with *weak integrability breaking* if the physical effects of integrability breaking do not manifest themselves at leading order in the coupling constant. For example, the relaxation time towards the Gibbs Ensemble of the non-integrable $H(\lambda)$ behaves as $t \sim \lambda^{-\kappa}$, with $\kappa \geq 4$ [18] as opposed to the standard case with $\kappa = 2$ [25, 26].

Weak integrability breaking as we defined it appears to be associated with *quasi-conserved quantities* [12, 16, 19]. There are also other mechanisms guaranteeing the persistence of certain dynamical features of integrability (for example, for the stability of

superdiffusion see [27]), but in the following we focus on the weak breaking associated with the quasi-conserved quantities.

If a certain Hamiltonian $H_0$ is integrable, then it possesses a family of conserved charges $Q_\alpha$, such that each charge is extensive with a local operator density, $H$ is a member of the family, and the charges commute among themselves:

$$[Q_\alpha, Q_\beta] = 0. \tag{14}$$

If the perturbation (10) was integrable, then each of the $Q_\alpha$ could be extended into a $\lambda$ dependent operator, and the commutativity could be preserved for all $\alpha$. For later use we introduce the notation $Q_\alpha^{(0)}$, which stands for the conserved charge in the original model at $\lambda = 0$.

We say that a certain charge $Q_\alpha^{(0)}$ is quasi-conserved, if the commutation with $H(\lambda)$ is preserved to the leading order in $\lambda$. In other words, if there is an operator $Q_\alpha^{(1)}$, which is extensive with a local density, such that

$$[H + \lambda V, Q_\alpha^{(0)} + \lambda Q_\alpha^{(1)}] = \mathcal{O}(\lambda^2), \tag{15}$$

which is equivalent to the relation

$$[V, Q_\alpha^{(0)}] + [H, Q_\alpha^{(1)}] = 0. \tag{16}$$

This relation should hold for the model with every length $L$ large enough that the charges can be defined there, or formally in the infinite volume limit.

At this point the main question is how to find solutions of (16). More precisely, for a given integrable model how to classify the perturbations $V$ which allow the solution of (16) for at least a subset of the original charges. If a solution is found, then we say that the combination $Q_\alpha + \lambda Q_\alpha^{(1)}$ is a *quasi-charge* for the given perturbation. Note that if a solution is found, then it is not unique, because $Q_\alpha^{(1)}$ can be modified by adding any operator which commutes with unperturbed Hamiltonian.

A formal solution to the problem can be given using the AGP itself. Let us assume for a moment that the model Hamiltonian is free of degeneracies, and the AGP can be defined unambiguously in the eigenbasis of $H$. This assumption is generally not true in translationally invariant integrable models. However, at this point we will sketch a formal procedure, therefore we neglect the issue of degeneracies for a moment.

Now consider the equation (5), which defines the AGP $\mathcal{A}_\lambda$ for any chosen perturbation $V$. We extend this relation to the higher charges, by making them $\lambda$-dependent and prescribing the evolution in $\lambda$ as

$$i\partial_\lambda Q_\alpha(\lambda) = [\mathcal{A}_\lambda, Q_\alpha(\lambda)] - i\mathcal{F}_\alpha(\lambda), \tag{17}$$

where the operator $\mathcal{F}_\alpha(\lambda)$ is again diagonal in the instantenous eigenbasis of $H(\lambda)$. Explicitly,

$$\mathcal{F}_\alpha(\lambda) = -\sum_n \frac{\partial Q_{\alpha,n}(\lambda)}{\partial \lambda} |n(\lambda)\rangle\langle n(\lambda)|. \tag{18}$$

Then it follows from the Jacobi identity for the triplet of operators $(H, Q_\alpha^{(0)}, \mathcal{A}_{\lambda=0})$ and from the commutativity

$$[H, Q_\alpha^{(0)}] = 0 \tag{19}$$

that the first order corrections in $\lambda$ satisfy (16). The problem with this formal approach is that it does not guarantee that the first order correction $Q_\alpha^{(1)}$ has the required locality properties, namely that it has a local or quasi-local operator density.

The problem of selecting "good" AGP operators for this purpose was considered in a seemingly unrelated part of the literature, namely in the theory of long range deformations, originally worked out for the spin chains describing scaling dimensions of local operators in the planar limit of $N = 4$ super-symmetric Yang-Mills theory [20, 21]. In the case of long range deformations every charge is expanded into a formal and infinite power series in $\lambda$:

$$Q_\alpha(\lambda) = \sum_{n=0}^{\infty} \frac{\lambda^n}{n!} Q_\alpha^{(n)}, \qquad (20)$$

such that the family of charges exactly preserves commutativity to each order in $\lambda$. Such long-range deformation can be used to build systems with weak integrability breaking, simply by truncating the above power series to the first or the first few terms [19].

The works [20, 21] considered the long range deformations in infinite volume, and the generating equations (5)-(17), focusing on the so-called fundamental models associated to the group $SU(N)$ with $N \geq 2$. The diagonal pieces $\mathcal{F}_\alpha(\lambda)$ were discarded, which is possible in the formal infinite volume limit. It was found that there are two types of generators which guarantee the locality of the deformations: those of the so-called boost and the bi-local type. For a review of these concrete cases we refer to the works [17, 19]. We should note that long range deformations of the bi-local type can be seen as a lattice version of the (generalized) $T\bar{T}$-deformations [28, 29], and this observation has been used in a series to construct and study new integrable models [30–34].

The connection between the AGP and the generator of long range deformations was observed already in [12, 19]. However, there are crucial differences too. The generator of the long range deformations was defined formally in an infinite volume setting, and its action was defined directly on the infinite volume asymptotic states, thus avoiding the problems of degeneracies. In contrast, the AGP is an inherently finite volume quantity, it is well defined only if the diagonal terms are also treated, and it should be compatible with all the degeneracies of the finite volume system. It can be attributed to these differences, that explicit formulas for the real space representation of the AGP are not known, not even in the cases of weak integrability breaking.

The norm of the AGP was investigated numerically in [12]. These numerical observations showed that the AGP undergoes a crossover from polynomial to exponential scaling as the size of the system grows. Moreover, the value of the crossover parameter itself exhibits exponential scaling with the system size, characterized by $\lambda_* = e^{-\gamma L}$, where $\gamma$ is a real positive number. This finding provides valuable insights into the behavior of the AGP in systems with weak integrability-breaking perturbations.

In the next section we are going to discuss models which are equivalent to free fermions, and we will consider two types of perturbations: those that are local and non-local with respect to the fermions, respectively. In the case of local perturbations we also find a crossover from polynomial to exponential scaling, even though the perturbation may break integrability strongly.

## 3 Weak integrability breaking of free fermions

In this section we examine free fermionic models with a special class of perturbations: we limit ourselves to 4-fermion perturbations, which are homogeneous in space and which conserve the fermion number. For these cases we show that it is always possible to build quasi-charges with the desired locality properties. Therefore, we establish weak integrability breaking. The corresponding AGP norm is considered afterwards in the next section.

## 3.1 Model and setup

As our main example we choose the so-called $XX$ model, which is defined as

$$H_{XX} = \frac{1}{4} \sum_j \left( X_j X_{j+1} + Y_j Y_{j+1} \right). \tag{21}$$

Here and in the following $X_j$, $Y_j$ and $Z_j$ are the standard Pauli operators acting at site $j$. We will consider this model with periodic boundary condition. In this case the summation over $j$ above runs from 1 to $L$ and periodicity is assumed for the indices. Here we apply the standard Jordan-Wigner transformation:

$$c_j^\dagger = \left( \prod_{k=1}^{j-1} Z_k \right) \sigma_j^-, \qquad c_j = \left( \prod_{k=1}^{j-1} Z_k \right) \sigma_j^+, \tag{22}$$

which gives us the fermionic model defined as

$$H_0 = \frac{1}{2} \sum_{j=1}^{L} \left( c_{j+1}^\dagger c_j + h.c. \right), \qquad c_{j+L} \equiv c_j. \tag{23}$$

This model is equivalent to the periodic $XX$ model only in the sector with an odd number of fermions: in the sector with an even number of fermions an extra factor of $-1$ appears in the boundary conditions for the fermions. However, this sign difference is not essential to our computations, therefore we dismiss it in the following, and in most of our examples we treat the model Hamiltonian (23) directly.

The Hamiltonian (23) can be diagonalized by standard methods. We use the Fourier transform of the fermionic operators:

$$c_p = \frac{1}{\sqrt{L}} \sum_{j=1}^{L} e^{ipj} c_j, \qquad c_p^\dagger = \frac{1}{\sqrt{L}} \sum_{j=1}^{L} e^{-ipj} c_j^\dagger, \tag{24}$$

to arrive at

$$H_0 = \sum_{n=1}^{L} \varepsilon_p c_p^\dagger c_p, \qquad p = \frac{2\pi n}{L}, \tag{25}$$

where the spectrum of quasi-particles is given by

$$\varepsilon_p = \cos(p). \tag{26}$$

The Jordan-Wigner transformation is highly non-local, and this can have consequences for the locality properties of the perturbations: operators which are local in the XX chain might not be local in the fermionic representation, and vice versa. The non-locality can arise from the Jordan-Wigner string $\prod_j Z_j$, therefore locality is preserved if this string of operators cancels. It follows, that those operators which can be represented by an even number of fermions close to each other are local on both sides of the duality. Let us now consider examples for such operators.

First we define the fermion number operators

$$n_j = c_j^\dagger c_j, \tag{27}$$

which can be represented in the original spin language as

$$n_j = \frac{1 - Z_j}{2}. \tag{28}$$

Any combinations of nearby $n_j$ operators are local both in the $XX$ chain and also in the fermionic chain.

In contrast, let us consider the local operators $X_j$. It follows from (22) that they can be expressed as

$$X_j = \left( \prod_{k=1}^{j-1} (1 - 2n_k) \right) (c_j + c_j^\dagger). \tag{29}$$

Any perturbation with an odd number of $X$-operators in the $XX$ chain is therefore highly non-local with respect to the fermions. It will be demonstrated later that this is crucial for the behaviour of the AGP norm.

### 3.2 Quasi-charges

Here we examine a Hamiltonian of the form (23). We consider the 4-fermion perturbation with $U(1)$ and translation symmetry:

$$H = H_0 + \lambda V, \tag{30}$$

where[1]

$$H_0 = \sum_p \cos(p) c_p^\dagger c_p \quad, \quad V = \frac{1}{L} \sum_{\vec{p} \in \mathcal{K}} V(p_1, p_2, p_3, p_4) c_{p_1}^\dagger c_{p_2}^\dagger c_{p_3} c_{p_4}. \tag{31}$$

Here the sum is over the discrete momenta $p_i = 2\pi n_i / L$, $n_i = 1, 2, ..., L$ and we defined the set:

$$\mathcal{K} \equiv \Big\{ p_i : \ p_1 + p_2 = p_3 + p_4 \pmod{2\pi} \Big\}, \tag{32}$$

where momentum conservation is the consequence of translational invariance. One can sum over momentum $p_4$ to rewrite the perturbation in the following form:

$$V = \frac{1}{L} \sum_{p_1, p_2, p_3} \tilde{V}(p_1, p_2, p_3) c_{p_1}^\dagger c_{p_2}^\dagger c_{p_3} c_{p_1 + p_2 - p_3}, \tag{33}$$

where we defined $\tilde{V}(p_1, p_2, p_3) \equiv V(p_1, p_2, p_3, p_1 + p_2 - p_3)$ and used the $2\pi$ periodicity of fermion operators $c_{p+2\pi n} \equiv c_p$.

At this point, we do not specify the concrete values of $V(\vec{p})$ or the real space representation of operator $V$. Instead, we just make the natural assumption that $V(\vec{p})$ is a continuous function of $p_i$, scaling $\mathcal{O}(1)$ with system size $L$ and bounded by constant:

$$\max_{\vec{p} \in \mathcal{K}} |V(\vec{p})| = \mathcal{V}. \tag{34}$$

Now we attempt to solve (16) for the selected charges $Q_\alpha^{(0)}$. In principle, we could choose the zeroth order charges to be any linear combination of the fermionic mode operators, i.e. $Q^{(0)} = \sum_k F(k) n(k)$, with arbitrary independent functions $F(k)$. However, in order to guarantee locality in coordinate space we consider the known family of local charges $Q_{(n,\sigma)}^{(0)}$ with $n \in \mathbb{N}$, $\sigma = 0, 1$, given by

$$Q_{(n,\sigma)}^{(0)} = \frac{1}{2}(-i)^\sigma \sum_{j=1}^L (c_j^\dagger c_{j+n} + (-1)^\sigma c_{j+n}^\dagger c_j). \tag{35}$$

---

[1]Throughout this subsections we use the shorthand $\vec{p}$ for the sequence of momenta $p_1, p_2, p_3, p_4$, and $\boldsymbol{p}$ for $p_1, p_2, p_3$.

These charges are expressed in Fourier space as

$$Q^{(0)}_{(n,\sigma)} = \sum_k \epsilon^{(n,\sigma)}(k) n(k) \tag{36}$$

with

$$\epsilon^{(n,0)}(k) = \cos(nk), \qquad \epsilon^{(n,1)}(k) = \sin(nk). \tag{37}$$

The Hamiltonian itself is a member of the family, as $H = Q^{(0)}_{(1,0)}$.

Let us now consider the family of charges with $\sigma = 0$. We attempt to find a solution to (16) directly in Fourier space. The first commutator in (16) is found to be

$$\left[ V, Q^{(0)}_{(n,0)} \right] = -\frac{1}{L} \sum_{\boldsymbol{p} \in \mathcal{K}} \left[ \cos(np_1) + \cos(np_2) - \cos(np_3) - \cos(np_4) \right] V(\vec{p}) c^\dagger_{p_1} c^\dagger_{p_2} c_{p_3} c_{p_4} =$$

$$-\frac{4}{L} \sum_{p_1, p_2, p_3} \cos\left( n\frac{p_1 + p_2}{2} \right) \sin\left( n\frac{p_3 - p_1}{2} \right) \sin\left( n\frac{p_3 - p_2}{2} \right) \tilde{V}(\boldsymbol{p}) c^\dagger_{p_1} c^\dagger_{p_2} c_{p_3} c_{p_1 + p_2 - p_3}. \tag{38}$$

Then a formal solution to the first-order correction is given by

$$\tilde{Q}^{(1)}_{(n,0)} = \frac{1}{L} \sum_{p_1, p_2, p_3} \tilde{f}_n(\boldsymbol{p}) \tilde{V}(\boldsymbol{p}) c^\dagger_{p_1} c^\dagger_{p_2} c_{p_3} c_{p_1 + p_2 - p_3}, \tag{39}$$

with

$$\tilde{f}_n(p_1, p_2, p_3) = \frac{\cos\left( n\frac{p_1 + p_2}{2} \right) \sin\left( n\frac{p_3 - p_1}{2} \right) \sin\left( n\frac{p_3 - p_2}{2} \right)}{\cos\left( \frac{p_1 + p_2}{2} \right) \sin\left( \frac{p_3 - p_1}{2} \right) \sin\left( \frac{p_3 - p_2}{2} \right)}. \tag{40}$$

We stress that this is just a formal result because both the numerator and the denominator can be zero. An actual solution is found only if all zeroes of the denominator are cancelled by the numerator.

Therefore, the necessary condition for solving (16) reads:

$$\cos\left( n\frac{p_1 + p_2}{2} \right) \sin\left( n\frac{p_3 - p_1}{2} \right) \sin\left( n\frac{p_3 - p_2}{2} \right) \tilde{V}(p_1, p_2, p_3) = 0 \ \text{ for } \ \boldsymbol{p} \in \mathcal{E}, \tag{41}$$

where we defined the set of momentum with zero energy excitation (given by denominator of (40)):

$$\mathcal{E} \equiv \left\{ p_i : \ \cos\left( \frac{p_1 + p_2}{2} \right) \sin\left( \frac{p_3 - p_1}{2} \right) \sin\left( \frac{p_3 - p_2}{2} \right) = 0 \right\}. \tag{42}$$

We argue that the condition (41) is satisfied for any function $\tilde{V}(\boldsymbol{p})$ if $n$ is odd. If for some $p^*$ we have $\sin(p^*) = 0$, then $\sin(n \, p^*) = 0$ for $n \in \mathbb{Z}$; if for some $p^*$ we have $\cos(p^*) = 0$, then $\cos(n \, p^*) = 0$ only for odd $n$. Thus, the necessary condition for the existence of a solution to (16) is satisfied for odd $n$ and any function $\tilde{V}(\boldsymbol{p})$.

As we already mentioned, one has the freedom of adding operators that commute with the initial Hamiltonian to the corrections of the charges. We add diagonal contributions to the charges (39) by analytical continuation of the function $\tilde{f}_n(\boldsymbol{p})$ to $\boldsymbol{p} \in \mathcal{E}$:

$$Q^{(1)}_{(n,0)} = \frac{1}{L} \sum_{p_1, p_2, p_3} f_n(\boldsymbol{p}) \tilde{V}(\boldsymbol{p}) c^\dagger_{p_1} c^\dagger_{p_2} c_{p_3} c_{p_1 + p_2 - p_3}, \qquad f_n(\boldsymbol{p}) \equiv \lim_{\tilde{\boldsymbol{p}} \to \boldsymbol{p}} \tilde{f}_n(\tilde{\boldsymbol{p}}). \tag{43}$$

The analytically continued function $f_n(p_1, p_2, p_3)$ is now smooth.

With this, we have constructed the quasi-charges for the 4-fermion perturbations of the $XX$ model. In the following subsection, we discuss the locality of the resulting operators.

However, first we comment here on higher perturbations, that are local in fermions. For simplicity, we focus on the 6-body perturbations. There the formal ratio of eigenvalue differences analogous to (40) takes the form

$$\frac{\cos(np_1) + \cos(np_2) + \cos(np_3) - \cos(np_4) - \cos(np_5) - \cos(np_6)}{\cos(p_1) + \cos(p_2) + \cos(p_3) - \cos(p_4) - \cos(p_5) - \cos(p_6)}, \qquad (44)$$

where momentum conservation $p_1 + p_2 + p_3 = p_4 + p_5 + p_6 \pmod{2\pi}$ holds due to translation invariance. In contrast to the 4-fermion case, there is now no way to factorize the expressions of cosines in the numerator and denominator of (44). This implies that submanifolds defined by the zeroes of the numerator and the denominator will generally not coincide, leading to divergences for the first-order correction $Q_{(n,0)}^{(1)}$. This implies that the quasi-charges can not be defined, and the 6-body perturbations are generally strongly breaking integrability.

## 3.3   Locality of quasi-charges

In this subsection we discuss locality properties of the results in equations (40)-(43). We focus on the case of $\tilde{V}(\boldsymbol{p})$ being polynomial in $e^{ip_j}$ of the following form:[2,3]

$$\tilde{V}(p_1, p_2, p_3) = \sum_{m_j=-M}^{M} v(m_1, m_2, m_3) e^{i(p_1 m_1 + p_2 m_2 - p_3 m_3)}, \qquad v(-\boldsymbol{m}) = v(\boldsymbol{m})^*, \qquad (45)$$

where $M$ (the highest order in $e^{ip_j}$) does not scale with system size $L$.

Such a form arises when the perturbation involves fermionic operators placed only $M$ sites apart. For instance, the perturbation operator $V = \sum_j n_j n_{j+l}$ (which is integrability breaking for $l \geq 2$ and integrability preserving for $l = 1$) in momentum space looks as:

$$V_l \equiv \sum_{j=1}^{L} n_j n_{j+l} = \frac{1}{L} \sum_{p_1, p_2, p_3} e^{i(p_2 - p_3)l} c_{p_1}^\dagger c_{p_2}^\dagger c_{p_3} c_{p_1+p_2-p_3}, \qquad (46)$$

so in this case the potential $\tilde{V}(p_1, p_2, p_3) = e^{i(p_2-p_3)l}$ is of the form (45) with highest order $M = l$.

The function $f_n(\boldsymbol{p})$ can also be rewritten as a polynomial of $e^{ip_j}$ using the simple formulas for the ratio of sines and cosines:

$$\frac{\sin(nx)}{\sin(x)} = \sum_{m=-\frac{(n-1)}{2}}^{\frac{(n-1)}{2}} e^{2ixm}, \quad \frac{\cos(nx)}{\cos(x)} = (-1)^{\frac{n-1}{2}} \sum_{m=-\frac{(n-1)}{2}}^{\frac{(n-1)}{2}} e^{2ixm}(-1)^m \quad \text{for odd } n. \quad (47)$$

Then the function

$$f_n(\boldsymbol{p}) = (-1)^{\frac{n-1}{2}} \sum_{m_j=-\frac{(n-1)}{2}}^{\frac{(n-1)}{2}} (-1)^{m_1} e^{ip_1(m_1-m_3)+ip_2(m_1-m_2)+ip_3(m_2+m_3)} \qquad (48)$$

is also a polynomial of the form (45) with the highest order in $e^{ip_j}$ equal to $(n-1)$.

---

[2]The negative sign convention $-m_3 p_3$ in the exponent leads to simpler formulas later. Compare this to the definition of $\boldsymbol{p} \cdot \boldsymbol{m}$ after (49).

[3]The summation over $m_j$ is a shorthand for three distinct summations over $m_1, m_2, m_3$ each having the same lower and upper bound.

The function $f_n(\boldsymbol{p})\tilde{V}(\boldsymbol{p})$ can also be written as a polynomial in $e^{ip_j}$:

$$f_n(\boldsymbol{p})\tilde{V}(\boldsymbol{p}) = \sum_{m_j=-(n-1+M)}^{(n-1+M)} \xi(\boldsymbol{m})e^{i\boldsymbol{p}\cdot\boldsymbol{m}}, \tag{49}$$

where $\xi(\boldsymbol{m}) = \xi(m_1, m_2, m_3)$ are coefficients, which do not scale with $L$, and we used the notation $\boldsymbol{p}\cdot\boldsymbol{m} = p_1 m_1 + p_2 m_2 - p_3 m_3$. Now the quasi-charges given by (43) can be represented in the following way:

$$Q_{(n,0)}^{(1)} = \sum_{m_j=-(n-1+M)}^{(n-1+M)} \xi(\boldsymbol{m})\frac{1}{L}\sum_{\boldsymbol{p}} e^{i\boldsymbol{p}\cdot\boldsymbol{m}}c_{p_1}^\dagger c_{p_2}^\dagger c_{p_3} c_{p_1+p_2-p_3}. \tag{50}$$

Notice that each term of the form

$$\chi(\boldsymbol{m}) = \frac{1}{L}\sum_{p_1,p_3,p_3} e^{i(p_1 m_1+p_2 m_2-p_3 m_3)}c_{p_1}^\dagger c_{p_2}^\dagger c_{p_3} c_{p_1+p_2-p_3} \tag{51}$$

is local in coordinate space. To show this let us transform the previous equation to coordinate space using (24):

$$\chi(\boldsymbol{m}) = \frac{1}{L^3}\sum_{x_j=1}^{L} c_{x_1}^\dagger c_{x_2}^\dagger c_{x_3} c_{x_4} \sum_{\boldsymbol{p}} e^{i\boldsymbol{p}\cdot\boldsymbol{m}-ip_1 x_1-ip_2 x_2+ip_3 x_3+i(p_1+p_2-p_3)x_4}. \tag{52}$$

Summing over components of $\boldsymbol{p}$, we obtain:

$$\chi(\boldsymbol{m}) = \sum_{x_i=1}^{L} c_{x_1}^\dagger c_{x_2}^\dagger c_{x_3} c_{x_4} \delta_{m_1-x_1+x_4}\delta_{m_2-x_2+x_4}\delta_{-m_3+x_3-x_4}. \tag{53}$$

Next, we sum over the coordinate components $x_1, x_2, x_3$, so we are left with the sum over one coordinate component:

$$\chi(\boldsymbol{m}) = \sum_{x=1}^{L} q_x(\boldsymbol{m}), \qquad q_x(\boldsymbol{m}) \equiv c_{x+m_1}^\dagger c_{x+m_2}^\dagger c_{x+m_3} c_x. \tag{54}$$

The density operator $q_x(\boldsymbol{m})$ has finite support $\mathrm{supp}(q_x(\boldsymbol{m})) \le 2(M+n-1)$ for bounded $|m_i| \le (n-1+M)$ from sum (50), and translation by $a$ sites acts as $\mathbb{T}_a q_x(\boldsymbol{m}) = q_{x+a}(\boldsymbol{m})$. So, the operator $\chi(\boldsymbol{m})$ is local.

Thus, the correction to charges given by a finite sum of local operators is also a local operator given by:

$$\boxed{Q_{(n,0)}^{(1)} = \sum_{m_j=-(n-1+M)}^{(n-1+M)} \xi(\boldsymbol{m})\chi(\boldsymbol{m}), \quad \chi(\boldsymbol{m}) = \sum_{x=1}^{L} c_{x+m_1}^\dagger c_{x+m_2}^\dagger c_{x+m_3} c_x.} \tag{55}$$

Above we constructed the charge corrections only for the $\sigma = 0$ family, and for odd $n$. In the same way, we can build the first-order corrections for the family of $\sigma = 1$ charges, but now for even $n$ (see Appendix B). Thus, half of the initial charges of $H_0$ can be deformed up to the first order in $\lambda$, which is the desired extensive number of corrections.

In this Subsection we focused on the perturbations described by $\tilde{V}(\boldsymbol{p})$ of the form (45). In Appendix C we also consider the general case of a bounded function $\tilde{V}(\boldsymbol{p})$, for which we prove the so-called quasi-locality and pseudo-locality of the resulting quasi-charge.

### 3.4 Weak breaking and long range deformations

In the previous Section 2.3 we explained that the long range deformations of integrable spin chains always lead to weak integrability breaking, simply by truncating the deformation to some chosen order in the deformation parameter. It is an interesting open question, whether all cases of weak integrability breaking can be obtained in such a way.

There are only two classes of systematic long range deformations known in the literature: the boost and the bi-local type [20, 21]. In the case of the XX model, the work [19] investigated various local perturbations, and found that the specific perturbation $V_2$ defined in (45) is not generated from a long range deformation of either the boost or the bi-local type. This would imply that $V_2$ is strongly breaking integrability. In contrast, we found that this specific perturbation weakly breaks integrability; explicit coordinate-space expressions for the first few local quasi-charges corresponding to this perturbation may be found in Appendix B.

Our results imply that either there are more types of long range deformations for free fermions, or that not all cases of weak breaking follow from the long range deformations. At present it is not clear, whether the perturbation by $V_2$ can be extended to an actual long range deformation. It might be that even though the 4-fermion terms are only weakly breaking integrability, there is no way to extend the quasi-charges beyond leading order. This question deserves further study.

## 4 AGP for free fermions

In this section, we provide a rather general argument, which shows that in free fermionic systems the AGP scales polynomially if the perturbing operator is local with respect to the fermions. We start with an analytical calculation of the AGP for 4-fermion perturbations with $U(1)$ symmetry and conservation of momenta. Additionally, we also consider the AGP for 6-fermion perturbations, demonstrating that the scaling remains polynomial, even though those perturbations are generally strongly breaking integrability.

In the case of the 4-fermion perturbation we start again with

$$V = \frac{1}{L} \sum_{\boldsymbol{p}} \tilde{V}(p_1, p_2, p_3) c_{p_1}^\dagger c_{p_2}^\dagger c_{p_3} c_{p_1+p_2-p_3}, \tag{56}$$

where we assume again that $\tilde{V}$ is a continuous function bounded by a constant

$$\max_{\boldsymbol{p}} |\tilde{V}(p_1, p_2, p_3)| = \mathcal{V}, \tag{57}$$

and $V$ is a local operator. Here we would like to use (9), in order to express AGP. To do so let us notice that the simple structure of Hamiltonian implies exact time dependence for the fermion modes

$$c_p(t) = e^{-i\varepsilon_p t} c_p, \qquad c_p^\dagger(t) = e^{i\varepsilon_p t} c_p^\dagger. \tag{58}$$

In order to find the AGP at the point $\lambda = 0$ we use the formula (9), which in the present case reads

$$\mathcal{A} = -\frac{1}{2} \int\limits_{-\infty}^{\infty} \text{sign}(t) e^{-\mu|t|} V(t), \tag{59}$$

where the time-evolved perturbation term is given by

$$V(t) = \frac{1}{L} \sum_{\boldsymbol{p}} e^{i\omega_{\boldsymbol{p}} t} \tilde{V}(p_1, p_2, p_3) c_{p_1}^\dagger c_{p_2}^\dagger c_{p_3} c_{p_1+p_2-p_3}, \tag{60}$$

and $\omega_{\boldsymbol{p}} = \varepsilon_{p_1} + \varepsilon_{p_2} - \varepsilon_{p_3} - \varepsilon_{p_1+p_2-p_3}$. A straightforward calculation leads to:

$$\mathcal{A} = -\frac{i}{L} \sum_{\boldsymbol{p}} \frac{\omega_{\boldsymbol{p}}}{\omega_{\boldsymbol{p}}^2 + \mu^2} \, \tilde{V}(p_1, p_2, p_3) c_{p_1}^\dagger c_{p_2}^\dagger c_{p_3} c_{p_1+p_2-p_3}. \tag{61}$$

Now it can be argued that the norm of the operator above always grows at most polynomially. On the one hand, note that the number of terms in the summand grows polynomially with the volume. The growth of each term is determined by the $\tilde{V}$ coefficients and the $\omega$-dependent pre-factors, where the $\tilde{V}$ pre-factor comes from the perturbation which was assumed to be local. If there are special algebraic relations between the $\varepsilon_p$, then the sum can be exactly zero; however, in such a case the pre-factor vanishes due to the regularisation we discussed. Otherwise, the minimal non-zero value decays with a power of $L$, due to the typical quantization conditions that determine the $\varepsilon_p$. Therefore, with any choice of the volume dependence of the regulator $\mu$, the pre-factor can only grow polynomially with the volume.

This argument does not depend on the number of fermions in the perturbing operator. In fact, the same argument can be repeated for 6-fermion or even higher body perturbations. The only crucial point is that the number of fermion operators has to be limited by a constant, i.e. the perturbation $V$ has to be local with respect to the fermions, in order to have a polynomial growth of the AGP. This is one of our main results.

Below we investigate this statement for the 4-fermion and 6-fermion perturbations. In the 4-fermion case we are able to derive an analytical bound for the AGP norm, whereas for the 6-fermion perturbation, we can find the scaling using numerical analysis.

## 4.1 Bound on the AGP

Let us consider the above $4-$fermion perturbation and calculate the bound on the scaling of the AGP norm. Using (61) the regularized norm of AGP is given by:

$$||\mathcal{A}||^2 = \frac{1}{2^L L^2} \sum_{\boldsymbol{p}, \boldsymbol{q}} \frac{\omega_{\boldsymbol{p}}^2}{(\omega_{\boldsymbol{p}}^2 + \mu^2)^2} \tilde{V}(\boldsymbol{p}) \tilde{V}(\boldsymbol{q}) \text{Tr} \left[ c_{p_1}^\dagger c_{p_2}^\dagger c_{p_3} c_{p_1+p_2-p_3} c_{q_1}^\dagger c_{q_2}^\dagger c_{q_3} c_{q_1+q_2-q_3} \right]. \tag{62}$$

Here we used that $\text{Tr}[c_{p_1}^\dagger c_{p_2}^\dagger c_{p_3} c_{p_1+p_2-p_3} c_{\tilde{p}_1}^\dagger c_{q_2}^\dagger c_{q_3} c_{q_1+q_2-q_3}] \neq 0$ only if $\omega_{\boldsymbol{p}} = -\omega_{\boldsymbol{q}}$, otherwise the diagonal matrix element of the operator expression inside the trace vanishes in the eigenbasis of $H_0$. Also, we take into account that the terms with $\omega_{\boldsymbol{p}} = 0$ do not contribute to the norm of the AGP due to the regularization discussed before.

In order to bound the norm of AGP, we first bound the nonzero energy difference $\omega_{\boldsymbol{p}}$ in sum (62) which reads

$$\omega_{\boldsymbol{p}} = \varepsilon_{p_1} + \varepsilon_{p_2} - \varepsilon_{p_3} - \varepsilon_{p_1+p_2-p_3} = 4 \cos\left(\frac{p_1+p_2}{2}\right) \sin\left(\frac{p_3-p_1}{2}\right) \sin\left(\frac{p_3-p_2}{2}\right). \tag{63}$$

In order to lower bound $|\omega_{\boldsymbol{p}}|$ notice that a trigonometric factor in the above formula takes up its minimal nonzero absolute value when its argument becomes as close to one of its zeros as possible - allowed by the $n_1, n_2, n_3$ momentum quantization numbers[4].

For the sine factors $\sin(\frac{n\pi}{L})$, where $n \in \mathbb{Z}$, the minimal nonzero value is $\sin(\frac{\pi}{L})$, whereas for the cosine $\cos(\frac{n\pi}{L})$ it is the same if $L$ is even, while becomes $\sin(\frac{\pi}{2L})$ for odd $L$. Both of these smallest values can be bounded from below simply by $1/L$ :

$$\sin\left(\frac{\pi}{L}\right) > \sin\left(\frac{\pi}{2L}\right) > \frac{1}{L} \qquad \text{for} \ \ L > 1. \tag{64}$$

---

[4]Note that these have also some constraints, since when $n_1 = n_2$ or $n_3 = n_4$, the string of fermionic operators vanishes inside the perturbation (31) as $c_{p_1}^\dagger c_{p_1}^\dagger = 0$ or $c_{p_3} c_{p_3} = 0$.

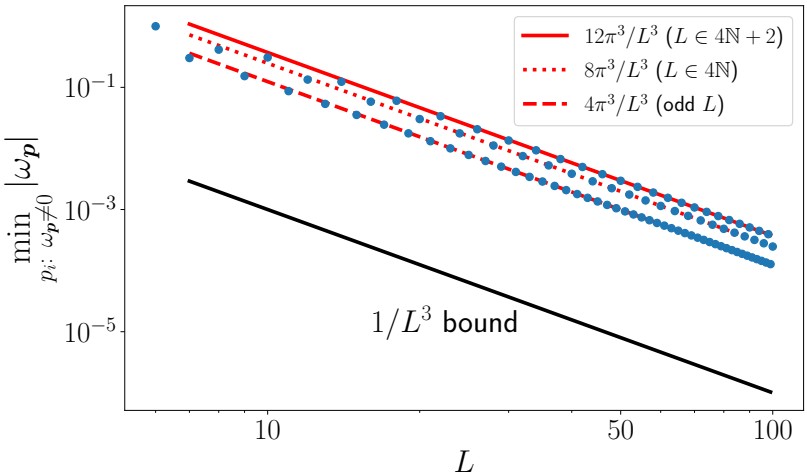

Figure 1: Scaling of minimal gap $\min_{\boldsymbol{p} \notin \mathcal{E}} |\omega_{\boldsymbol{p}}|$ for 4-fermion interaction with the system size $L$. The gap scales polynomially with the number of sites $\sim L^{-3}$. The bound is given by (65).

Thus, for the minimal nonzero energy difference given by (63), we have

$$\min_{\boldsymbol{p} \notin \mathcal{E}} |\omega_{\boldsymbol{p}}| > 1/L^3, \quad L > 1, \tag{65}$$

where $\mathcal{E}$ is set of momentum with zero energy difference $\omega_{\boldsymbol{p}} = 0$ introduced before in (42). We can also verify the obtained lower bound by numerically calculating the minimum energy difference. We use (63) and determine the minimum energy difference as a function of the system size see Figure 1.

Now we can calculate the bound on the AGP itself. Using the $\omega_{\boldsymbol{p}}$ bound (65) we deduce that:

$$\frac{\omega_{\boldsymbol{p}}^2}{(\omega_{\boldsymbol{p}}^2 + \mu^2)^2} < L^6. \tag{66}$$

Therefore the only thing we need to bound now is the sum over momenta of the coefficients $\tilde{V}(\boldsymbol{p})$:

$$\left| \sum_{\boldsymbol{p},\boldsymbol{q}} \tilde{V}(\boldsymbol{p}) \tilde{V}(\boldsymbol{q}) \text{Tr} \left[ c_{p_1}^\dagger c_{p_2}^\dagger c_{p_3} c_{p_1+p_2-p_3} c_{q_1}^\dagger c_{q_2}^\dagger c_{q_3} c_{q_1+q_2-q_3} \right] \right| < 4! \, \mathcal{V}^2 \, L^3 2^{L-4}. \tag{67}$$

Here we used some simple statements about the traces of the fermionic operators, with details given in Appendix A.

Thus, the AGP is bounded by a polynomial function of the system size

$$\boxed{||\mathcal{A}||^2 < \frac{3}{2} \mathcal{V}^2 L^7.} \tag{68}$$

To further demonstrate the polynomial scaling of the norm, we take a specific example and calculate the AGP numerically for the system defined by the following perturbation:

$$V = \sum_{j=1}^{L} n_j n_{j+2}. \tag{69}$$

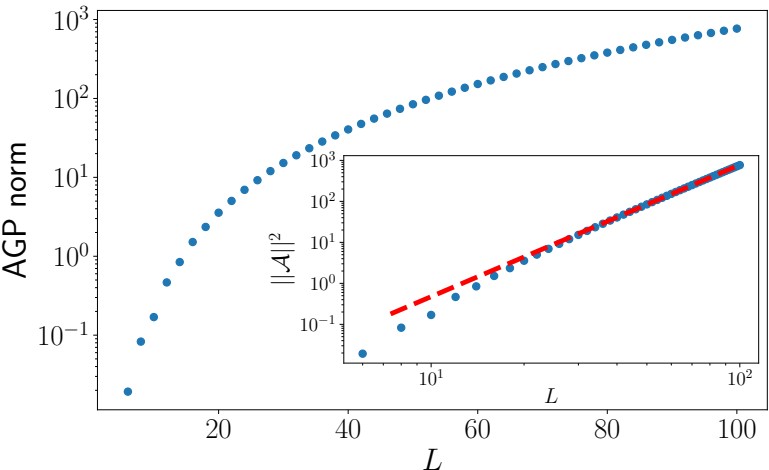

Figure 2: Scaling of the AGP norm $||\mathcal{A}||^2$ with the system size for the integrability breaking perturbation $V_2 = \sum_j n_j n_{j+2}$. The regularization is given by $\mu = L 2^{-L}$. The inset shows the scaling of the AGP norm on a log-log scale. It scales polynomially with the system size $||\mathcal{A}||^2 \sim L^\alpha$, where $\alpha \approx 3.2$ .

In Figure 2 we show the AGP's behaviour as a function of the system size, and we find that indeed it scales as some power of $L$. The exponent that we obtained numerically is smaller than the one appearing in the bound (68).

Having discussed the 4-fermion perturbations, now we turn our attention to the $6-$fermion perturbations[5]

$$V = \frac{1}{L^2} \sum_{\boldsymbol{p}} \tilde{V}(p_1, p_2, p_3, p_4, p_5) c_{p_1}^\dagger c_{p_2}^\dagger c_{p_3}^\dagger c_{p_4} c_{p_5} c_{\Delta p}, \tag{70}$$

where $\Delta p = p_1 + p_2 + p_3 - p_4 - p_5$, and we introduced the function $\tilde{V}(\boldsymbol{p})$ with 5 arguments in a similar fashion as we did in (56). In this case the AGP takes the form

$$||\mathcal{A}||^2 = \frac{1}{2^L L^4} \sum_{\boldsymbol{p}, \boldsymbol{q}} \frac{\omega_{\boldsymbol{p}}^2}{(\omega_{\boldsymbol{p}}^2 + \mu^2)^2} \tilde{V}(\boldsymbol{p}) \tilde{V}(\boldsymbol{q}) \text{Tr} \left[ c_{p_1}^\dagger c_{p_2}^\dagger c_{p_3}^\dagger c_{p_4} c_{p_5} c_{\Delta p} c_{q_1}^\dagger c_{q_2}^\dagger c_{q_3}^\dagger c_{q_4} c_{q_5} c_{\Delta q} \right]. \tag{71}$$

To determine the scaling of the AGP norm, we investigate the scaling of the minimal energy gap, given by:

$$\omega_{\boldsymbol{p}} = \cos(p_1) + \cos(p_2) + \cos(p_3) - \cos(p_4) - \cos(p_5) - \cos(\Delta p). \tag{72}$$

We argue that the minimal non-zero value scales polynomially with $1/L$. Every momentum $p_j$ appearing in the cosines has a specially selected value $2\pi I_j/L$, with some $I_j = 1, 2, \ldots, L$. These values come from the quantization conditions of the free model. Sometimes there can be coincidences when the whole sum is actually zero, and we assume that the smallest non-zero values appear close to these special coincidences. Then simply an expansion into a Taylor series around the zero values will tell us that the smallest non-zero values behave polynomially with $1/L$. As we increase the number of the terms in the sum, then by special cancellations it is possible to obtain higher and higher order cancellations in $1/L$, but we can never achieve an exponentially small non-zero value.

This argument is not rigorous. We searched the mathematical literature for a proof of our claim. The related problem of finding the minimum non-zero value of sums of roots

---

[5] Here $\boldsymbol{p}$ means the sequence $p_1, p_2, p_3, p_4, p_5$.

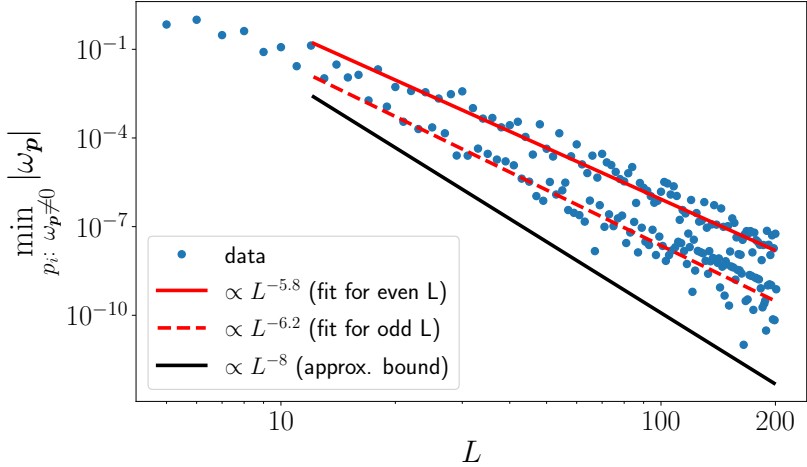

Figure 3: Polynomial scaling of the minimal energy difference $|\omega_p|$ for 6-fermion interactions. The numerical data is fitted for even and odd $L$ values separately, and the black line corresponds to the estimated bound $\propto L^{-8}$.

of unity appeared in the recent mathematical work [35], but for the case of five or more roots no proof was found. We did not find any other mathematical work discussing this question, thus it appears that this particular problem is not yet solved.

In order to substantiate our claim, we also performed further numerical studies, in the special case of the 6-body perturbations. In Figure 3 we plot the minimal energy among all possible values of $\boldsymbol{p}$, and we observe that the scaling of the minimal energy remains polynomial, specifically $\min\limits_{p_i:\ \omega_{\boldsymbol{p}} \neq 0} |\omega_{\boldsymbol{p}}| \sim L^{-(6.0 \pm 0.2)}$ when fitted. There seem to be points not respecting this scaling and thus the lower bound is better approximated by some other power

$$\min_{p_i:\ \omega_{\boldsymbol{p}} \neq 0} |\omega_{\boldsymbol{p}}| \gtrsim \text{const.} \times L^{-\beta}, \tag{73}$$

where $\beta \approx 8$ based on Figure 3. This allows us to estimate the scaling of the AGP using the formula in (71), resulting in:

$$||\mathcal{A}||^2 \lesssim \text{const.} \times L^{2\beta+1}, \tag{74}$$

that is, the scaling is bounded by $L^{17}$ from our numerics up to $L = 200$. It is likely that this high power could be lowered by more accurate estimates. However, for our purpose, it is enough that we establish the polynomial growth.

## 4.2 AGP at $\lambda \neq 0$

Until now, our focus has been on the behavior of the AGP at the integrable point $\lambda = 0$. We now shift our attention to the case of finite $\lambda$, employing the formulas given by (8) and (11).

At first, we consider the specific example given by (69) again. As discussed in Subsection 3.4, we expect this to be weak integrability breaking. We carry out the numerical analysis of the AGP norm, presenting its variation as a function of the system size $L$, where $7 \leq L \leq 19$ across different parameter values of $\lambda$. The results are shown in Figure 4. We observe a shift from polynomial to exponential scaling. This pattern is consistent with previously investigated examples of weak integrability breaking.

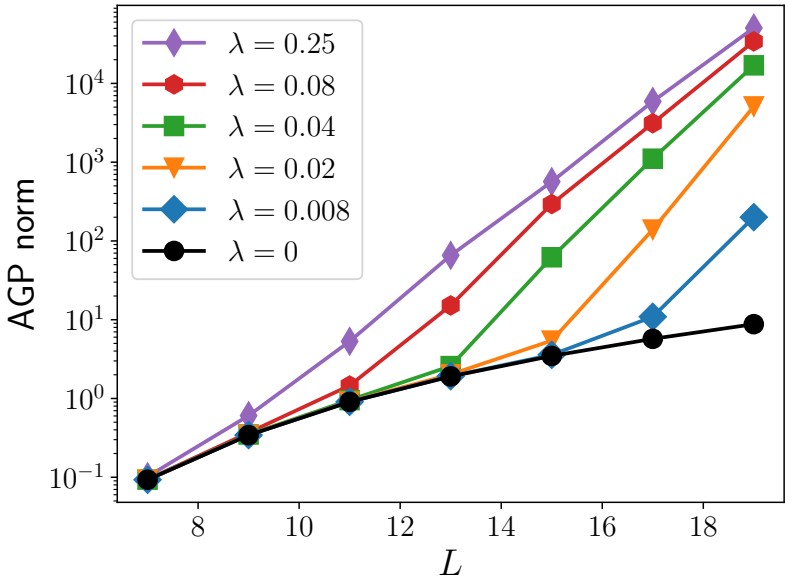

Figure 4: The AGP norm as a function of system size $L$, for the Hamiltonian of free fermions perturbed by the potential (69) for different values of the integrability breaking coupling $\lambda$. We observe the crossover from polynomial to exponential scaling.

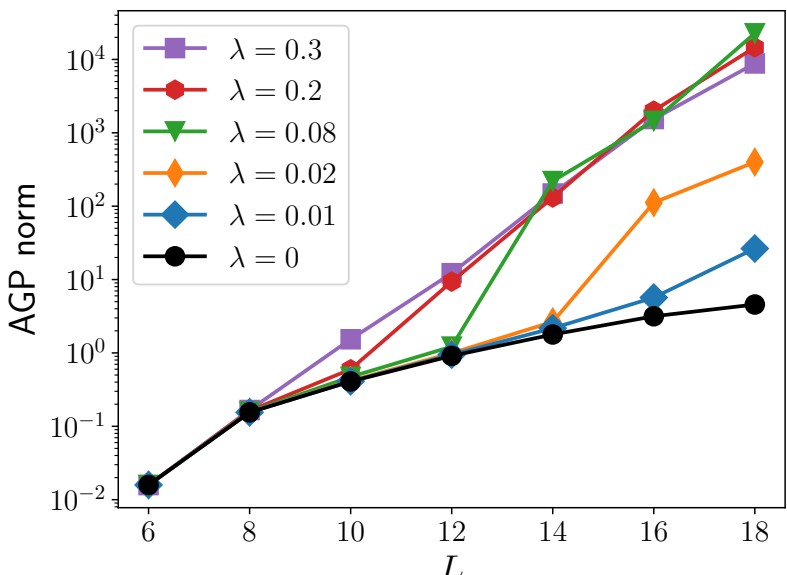

Figure 5: The AGP norm as a function of the system size, for the perturbation (75), plotted for several $\lambda$ values.

Now, we turn to the case of a 6-fermion perturbation

$$V = \sum_{j=1}^{L} n_j n_{j+1} n_{j+2}, \tag{75}$$

which is an example of strong integrability breaking operators by our argument given at the end of Subsection 3.2. As shown in Figure 5, there is a crossover from polynomial to exponential scaling, and it happens for smaller system sizes with increasing $\lambda$, similarly to the weak integrability breaking scenario. This indicates that the $\lambda \neq 0$ AGP seems to

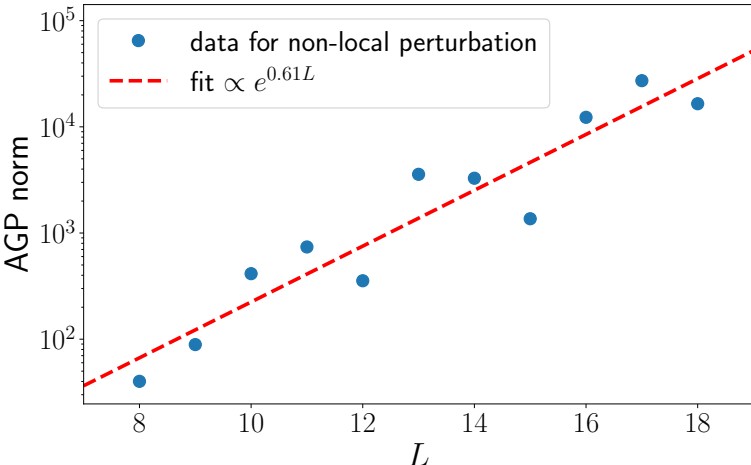

Figure 6: The AGP norm at $\lambda = 0$ in case of the $XX$ model for the perturbing operator $\sum_{j=1}^{L} X_j$. This operator breaks integrability, and it is non-local in the fermions, therefore we find an exponential scaling of the AGP norm.

be insensitive to whether we perturb our free fermionic model weakly or strongly. This is opposite to what has been found when perturbing interacting integrable models in [12].

### 4.3   AGP for non-local perturbations

In this subsection, we explore a perturbation that is non-local in the fermions. In this scenario, we expect exponential scaling with respect to the volume.

The Hamiltonian we consider is expressed in terms of the spin operators as

$$H = H_{XX} + \sum_{j=1}^{L} X_j, \tag{76}$$

where $H_{XX}$ is a Hamiltonian of $XX$ model we discussed in Subsection 3.1, and $X_j$ are corresponding Pauli matrices. Notice that now the perturbation term $V = \sum_j X_j$ is non-local in fermions because the $X_j$-s contain the Jordan-Wigner strings.

We use exact diagonalisation in order to calculate the AGP norm as a function of the system size; results are shown in Figure 6. We see that the scaling of adiabatic gauge potential is exponential with the system size $||A||^2 \sim e^{\kappa L}$, where $\kappa \sim 0.61(7)$.

## 5   Conclusions

In this work we investigated integrability breaking in systems solvable by free fermions. We obtained two main results, which escaped previous works. First, we proved that local 4-body perturbations are always weakly breaking integrability. Second, we found that the AGP norm scales polynomially for integrability breaking terms which are local with respect to the fermions, irrespective whether the integrability breaking is strong or weak (in the sense described in the main text). We focused on the specific example of the XX model, but all our arguments depend only on the free fermionic nature, and not on the other concrete details of the models.

Our first result might not be surprising from the point of view of Generalized Hydro-dynamics (GHD). It can be argued, that a particle number conserving 4-body interaction

does not give first order corrections to the hydrodynamical flow equations, simply as a result of energy and momentum conservation in the 2 body scattering events [36, 37]. However, in the present work we obtained the result by exact computations involving the fermionic fields.

In the concrete case of the XX model we found a particular 4-fermion perturbation, for which it is known that it does not follow from a known long range deformation, see the discussion in Section 3.4. This implies that either there are new types of long range deformations (at least for the free models), or that the possibilities for weak integrability breaking are simply just wider than the families of long range deformations.

Some of our results are based on mathematical statements about the minimal non-zero values of finite sums of roots of unity. These statements seem natural, but in fact they haven't been proven. A special case of such questions has been investigated recently in [35]. It would be useful to actually prove these statements.

# 6 Acknowledgements

We thank Jacopo De Nardis, Tomaž Prosen, Lenart Zadnik, Enej Ilievski and Márton Mestyán for useful discussions. This work was supported by the Hungarian National Research, Development and Innovation Office, NKFIH Grant No. K-145904 and the NK-FIH excellence grant TKP2021-NKTA-64. This work has been partially funded by the ERC Starting Grant 101042293 (HEPIQ) (A.T.). The work of RS was supported by the Program P1-0402 of Slovenian Research and Innovation Agency (ARIS).

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

## A  Trace identities

In this section, we provide useful formulas for traces of multiple spinless fermion operators. We will consider an equal number of creation and annihilation operators since we consider models with $U(1)$ global charge. The expression we evaluate is

$$\mathcal{I}^{(n)}_{i,j,k,l} = \text{Tr}\left[ c^\dagger_{i_1}...c^\dagger_{i_n} c_{j_1}...c_{j_n} c^\dagger_{k_1}...c^\dagger_{k_n} c_{l_1}...c_{l_n} \right]. \tag{77}$$

The simplest example we need is $n = 1$

$$\text{Tr}[c^\dagger_i c_j c^\dagger_k c_l] = \left( \delta^j_i \delta^l_k + \delta^l_i \delta^j_k \right) \text{Tr}\left[ n_i n_k \right] = 2^{L-2} \left( \delta^j_i \delta^l_k + \delta^l_i \delta^j_k \right). \tag{78}$$

The general expression is given by the sum over all different contractions of creation and annihilation operators, namely:

$$\mathcal{I}^{(n)}_{i,j,k,l} = 2^{L-2n} \sum_\sigma (-1)^{\#(\sigma)} \delta^{\sigma(j_1)}_{i_1}...\delta^{\sigma(j_n)}_{i_n} \cdot \delta^{\sigma(l_1)}_{k_1}...\delta^{\sigma(l_n)}_{k_n}, \tag{79}$$

where $\#(\sigma)$ is number of intersections of contraction. Notice that the number of terms in (79) is $(2n)!$.

## B  Examples for local quasi-charges

As stated at the end of Subsection 3.2 it is possible to build the $\sigma = 1$ quasi-charges as well for a 4-fermion perturbation, at least for even $n$. The calculation is very similar, here we only point out the differences. The trigonometric identities used in (38) now give

$$\begin{aligned}\sin(np_1) + \sin(np_2) &- \sin(np_3) - \sin(np_4) \\ &= 4\sin\left(n\frac{p_1+p_2}{2}\right)\sin\left(n\frac{p_3-p_1}{2}\right)\sin\left(n\frac{p_3-p_2}{2}\right),\end{aligned} \tag{80}$$

if $p_1 + p_2 = p_3 + p_4 \pmod{2\pi}$, thus in place of (47) we need to use

$$\frac{\sin(nx)}{\sin(x)} = \sum_{m=-\frac{n}{2}}^{\frac{n}{2}-1} e^{(2m+1)ix}, \quad \frac{\sin(nx)}{\cos(x)} = i(-1)^{n/2} \sum_{m=-\frac{n}{2}}^{\frac{n}{2}-1} e^{(2m+1)ix}(-1)^m \quad \text{for even } n. \tag{81}$$

After these steps, the machinery is very similar to the $\sigma = 0$ case, therefore let us here only summarize the final result for both families:

$$Q^{(1)}_{(n,\sigma)} = \sum_{m_j=-(n-1+M)}^{n-1+M} \xi_{(n,\sigma)}(\boldsymbol{m})\chi(\boldsymbol{m}), \quad \xi_{(n,\sigma)}(\boldsymbol{m}) = \sum_{m'_j=-M}^{M} X_{(n,\sigma)}(\boldsymbol{m}-\boldsymbol{m}')v(\boldsymbol{m}'), \tag{82}$$

where the coordinate-space coefficients are determined by the function

$$X_{(n,\sigma)}(\boldsymbol{m}) = i^\sigma(-1)^{\frac{n-1+\sigma}{2}}(-1)^{\frac{m_1+m_2-m_3-\sigma}{2}}\Theta_{(n,\sigma)}(m_1 + m_2 - m_3)$$
$$\times\,\Theta_{(n,\sigma)}(m_1 - m_2 - m_3)\Theta_{(n,\sigma)}(-m_1 + m_2 - m_3), \tag{83}$$

and where the characteristic function restricting the domain is

$$\Theta_{(n,\sigma)}(m) = \begin{cases} 1 & \text{if} \quad -n+1 \le m \le n-1 \quad \text{and} \quad m \in 2\mathbb{Z} + \sigma \\ 0 & \text{otherwise.} \end{cases} \tag{84}$$

For the specific family of perturbations (46), especially for $l = 1, 2$ we show explicitly the coordinate-space expression (82) of the first few constructed quasi-charges.

The (integrability preserving) perturbation $V_1 = \sum_j n_j n_{j+1}$ which is the analog of the $XXZ$-type perturbation $\sum_j Z_j Z_{j+1}$ for the $XX$ model, modifies the first non-trivial charge by the correction

$$Q^{(1)}_{(2,1)} = -i\left[\chi(2,0,1) + \chi(2,1,2)\right] + h.c., \tag{85}$$

which is exactly the $\mathcal{O}(\lambda)$ part of the corresponding $XXZ$ charge (see $H_3$ in equation (5.12) of [38]) when represented via fermions after Jordan-Wigner transformation. The same charge correction for the range-3 perturbation $V_2 = \sum_j n_j n_{j+2}$ looks as:

$$Q^{(1)}_{(2,1)} = -i\left[\chi(2,0,1) + \chi(2,1,2) + \chi(3,0,2) + \chi(3,1,3)\right] + h.c. \tag{86}$$

The next charge, i.e. $Q^{(0)}_{(3,0)}$ for which the construction is possible then gets its correction as

$$Q^{(1)}_{(3,0)} = \frac{1}{2}\chi(1,0,1) - \chi(2,0,2) - \chi(2,1,1)$$
$$+ \chi(3,0,1) + \chi(3,1,2) - \chi(2,1,3) + \chi(3,2,3) + h.c. \tag{87}$$

in case of the XXZ-type perturbation $V_1$, and as

$$Q^{(1)}_{(3,0)} = -\frac{1}{2}\left(\chi(1,0,1) + \chi(2,0,2) + \chi(3,0,3)\right) + \chi(2,1,1)$$
$$+ \chi(3,0,1) + \chi(3,1,2) - \chi(2,1,3) + \chi(3,2,3)$$
$$+ \chi(4,1,2) + \chi(4,1,3) - \chi(3,1,4) + \chi(4,2,4) + h.c. \tag{88}$$

for perturbation $V_2$.

## C  Locality of quasi-charges for general potential

In this section we consider the locality properties of quasi-charges given by equations (40)-(43) for different classes of functions $\tilde{V}(\boldsymbol{p})$. First, we define the notion of pseudolocality and quasilocality of an operator. Following the literature we define the Hilbert–Schmidt (HS) inner product and the corresponding HS norm of operators:

$$\langle A, B \rangle = \frac{1}{\mathcal{D}}\text{Tr}(A^\dagger B) - \frac{1}{\mathcal{D}^2}\text{Tr}(A^\dagger)\text{Tr}(B), \quad \|A\|^2_{\text{HS}} = \langle A, A \rangle, \tag{89}$$

where $\mathcal{D} = 2^L$ is Hilbert-space dimension. The operator $\mathcal{O}$ is *pseudolocal* if it satisfies the following two conditions:

$$1) \quad 0 < \lim_{L \to \infty} \frac{||\mathcal{O}||_{\mathrm{HS}}^2}{L} < +\infty \quad 2) \; \exists \; \text{local } b : \lim_{L \to \infty} \langle b, \mathcal{O} \rangle \neq 0. \tag{90}$$

The quasilocality property also constrains the local densities of operators. Namely, consider the translationally invariant operator $\mathcal{O}$ given by:

$$\mathcal{O} = \sum_{x=1}^{L-1} \mathcal{P}^x(q), \qquad q = \sum_{r=1}^{L} q_{[1,r]} \tag{91}$$

where $\mathcal{P}^x$ is translation operator by $x$ sites, $q$ is density and $q_{[1,r]}$ is local density supported over sites $1, ..., r$. Then, the operator $\mathcal{O}$ is *quasilocal* if it has exponentially decaying HS norm of local densities with increasing support:

$$\exists \; C, \xi > 0 : \quad ||q_{[1,r]}||_{\mathrm{HS}}^2 < C e^{-\xi r}. \tag{92}$$

Notice that a quasilocal operator is automatically pseudolocal.

**Pseudolocality**. Here we consider the general case of the function $\tilde{V}(\boldsymbol{p})$, which is bounded $\max_{\boldsymbol{p}} \tilde{V}(\boldsymbol{p}) = \mathcal{V}$, but not necessarily continuous in thermodynamic limit (e.g. disorder interaction). We show that in this case, the charge corrections are pseudolocal. The existence of an extensive number of pseudolocal charges constrains the transport properties of a quantum many-body system, in particular, it leads to ballistic scaling of the dynamical response function (see e.g. [39]).

Notice, that since we consider the quasi charges of the form $Q = Q^{(0)} + \lambda Q^{(1)} + \mathcal{O}(\lambda^2)$ the second condition is automatically satisfied due to the locality of the non-perturbed charges given by (35). In order to bound the HS norm of corrections we found (43), first we have to bound the function $f_n(p_1, p_2, p_3)$ given by Eq.(40). It's easy to prove by induction that:

$$\left| \frac{\cos(nx)}{\cos(x)} \right| \leq n \qquad \forall x : \quad \cos(x) \neq 0 \text{ and odd } n, \tag{93a}$$

$$\left| \frac{\sin(nx)}{\sin(x)} \right| \leq n \qquad \forall x : \quad \sin(x) \neq 0 \text{ and } \forall n \in \mathbb{Z}. \tag{93b}$$

The function $f_n(p_1, p_2, p_3)$ given by 3 ratios of cosines and sines is bounded by

$$|f_n(p_1, p_2, p_3)| \leq n^3, \qquad p_i \in \frac{2\pi}{L} \mathbb{Z}. \tag{94}$$

Let us calculate the operator norm of the first-order corrections:

$$||Q_{(n,0)}^{(1)}||_{\mathrm{HS}}^2 = \frac{1}{2^L L^2} \sum_{\boldsymbol{p}, \boldsymbol{q}} f_n(\boldsymbol{p}) f_n(\boldsymbol{q}) \tilde{V}(\boldsymbol{p}) \tilde{V}(\boldsymbol{q}) \mathrm{Tr} \left[ c_{p_1}^\dagger c_{p_2}^\dagger c_{p_3} c_{p_1+p_2-p_3} c_{q_1}^\dagger c_{q_2}^\dagger c_{q_3} c_{q_1+q_2-q_3} \right] \leq$$

$$\frac{\mathcal{V}^2 n^6}{2^L L^2} \sum_{\boldsymbol{p}, \boldsymbol{q}} \left| \mathrm{Tr} \left[ c_{p_1}^\dagger c_{p_2}^\dagger c_{p_3} c_{p_1+p_2-p_3} c_{q_1}^\dagger c_{q_2}^\dagger c_{q_3} c_{q_1+q_2-q_3} \right] \right| \leq \frac{3}{2} \mathcal{V}^2 n^6 L, \tag{95}$$

where we used (79) to decompose the trace of 8 fermionic operators as a sum of 4! Kronecker delta terms with overall multiplication factor $2^{L-4}$ (see Appendix A). From (95) follows that $\lim_{L \to \infty} ||Q_{(n,0)}^{(1)}||_{\mathrm{HS}}^2 / L < +\infty$, which implies $\lim_{L \to \infty} \left| \left\langle Q_{(n,0)}^{(0)}, Q_{(n,0)}^{(1)} \right\rangle \right| / L < +\infty$ from Cauchy–Schwarz inequality. Thus, we proved the pseudolocality of quasi-charges in the case of any bounded two-body interaction potential $\tilde{V}(\boldsymbol{p})$:

$$\boxed{\text{for odd } n : \quad 0 < \lim_{L \to \infty} \frac{||Q_{(n,0)}^{(0)} + \lambda Q_{(n,0)}^{(1)}||_{\mathrm{HS}}^2}{L} < +\infty}. \tag{96}$$

***Quasilocality***. Here we consider the case of smooth function $\tilde{V}(\boldsymbol{p})$. Let us transform the expression for charge corrections (43) to coordinate space:

$$Q_{(n,0)}^{(1)} = \sum_{x_1,x_2,x_3,x_4} F(x_1 - x_4, x_2 - x_4, x_4 - x_3) c_{x_1}^{\dagger} c_{x_2}^{\dagger} c_{x_3} c_{x_4}, \qquad (97)$$

where we defined[6]

$$F(y_1, y_2, y_3) \equiv \frac{1}{L^3} \sum_{\boldsymbol{p}} e^{-i\boldsymbol{p}\cdot\boldsymbol{y}} \tilde{V}(p_1, p_2, p_3) f_n(p_1, p_2, p_3). \qquad (98)$$

In the thermodynamic limit $L \to \infty$, $F(\boldsymbol{y})$ becomes the Fourier transform of a smooth function: $F(\boldsymbol{y}) \to \frac{1}{(2\pi)^3} \int d^3\boldsymbol{p}\ e^{-i\boldsymbol{p}\cdot\boldsymbol{y}} \tilde{V}(\boldsymbol{p}) f_n(\boldsymbol{p})$. The Fourier transform of a smooth function is exponentially decaying, namely $\exists A, \kappa :\ F(\boldsymbol{y}) \leq A e^{-\kappa|\boldsymbol{y}|}$ (Payley-Wiener theorem, see e.g. [40,41]). Thus, the density of quasi-charges (97) is exponentially localized, so they are quasilocal.

---

[6]In contrast to the definition of $\boldsymbol{p} \cdot \boldsymbol{m}$ after (49) here $\boldsymbol{p} \cdot \boldsymbol{y} = p_1 y_1 + p_2 y_2 + p_3 y_3$.