# Peer review of "Adiabatic gauge potential and integrability breaking with free fermions"

_SciPost Physics_

## Round 2 · Referee Report · Anonymous (Referee 1) · 2024-5-30

Report

Recently the scaling behavior of the adiabatic gauge potential has been introduced as a new measure for quantifying and defining quantum chaos. Specifically, according to current literature results one expects exponential growth of the AGP norm with volume for strong integrability breaking. The present manuscript gives analytical bounds for strongly integrability breaking local perturbations in a fermion system which show that the growth of the AGP norm is only polynomial, contradicting these claims in the literature. The analysis is backed up by compelling numerical data.

In summary the manuscript provides an interesting results on a very current much debated topic. I support publication in its current form.

Recommendation

Publish (easily meets expectations and criteria for this Journal; among top 50%)

---

## Round 2 · Referee Report · Anonymous (Referee 2) · 2024-6-28

Report

Report on the manuscript entitled Adiabatic gauge potential and integrability breaking with free fermions by Balázs Pozsgay, Rustem Sharipov, Anastasiia Tiutiakina and István Vona.

The Authors say to incorporate a new scenario to the already known exponential or polynomial scaling with system size of the norm of so-called adiabatic gauge potential (AGP) for chaotic or integrable many-body quantum systems, respectively. Namely, for free fermionic models subjected to local (with respect to fermions) weak or strong integrability breaking perturbations, the AGP norm scales polynomially when λ is set to zero. The weak and strong local perturbations are represented by four and six fermions terms respectively. For the former kind of perturbation, the Authors explicitely proved that indeed they are “always weakly breaking integrability”. Their staments are supported by numerical results on the integrable XX model. However, for finite lambda departing from zero, the Authors also find a crossover to the exponential scaling for large enough λ.

In general, I do find the manuscript well written, technically sound and timely. I consider this work as a good contribution to the theory of many-body quantum chaos. I feel comfortable suggesting publication in SciPost, however, before this happens I would like to ask the Authors to clarify something that could seem trivial but it is confusing me.

The picture they draw departs from considering a λ-dependent Hamiltonian H(λ)=H0+λV, where in particular H0= XX. To my understanding, for λ=0 the AGP norm scales polynomially with system size, just as it should be expected for an integrable model, the XX model. Once \lambda increases from zero the exponential scaling with system size is eventually recovered, just as it should be for chaotic systems. The crossover from polynomial to exponential scaling of the AGP norm for intermediate values of λ should be also expected, since this kind of behavior is well known in the many-body quantum chaos literature, say, for instance, from a level spacing statistics perspective. What does mean λ=0 in the context of the AGP not of the Hamiltonian? Please, could you clarify if I missunderstood?

Recommendation

Publish (meets expectations and criteria for this Journal)

---

## Editorial Decision

resubmitted